# Nanocomposite electrodes for high current density over 3 A cm$^{-2}$ in solid oxide electrolysis cells

Hiroyuki Shimada [1]*, Toshiaki Yamaguchi[1], Haruo Kishimoto[2], Hirofumi Sumi [1], Yuki Yamaguchi[1], Katsuhiro Nomura[1] & Yoshinobu Fujishiro [1]

Solid oxide electrolysis cells can theoretically achieve high energy-conversion efficiency, but current density must be further increased to improve the hydrogen production rate, which is essential to realize widespread application. Here, we report a structure technology for solid oxide electrolysis cells to achieve a current density higher than 3 A cm$^{-2}$, which exceeds that of state-of-the-art electrolyzers. Bimodal-structured nanocomposite oxygen electrodes are developed where nanometer-scale $Sm_{0.5}Sr_{0.5}CoO_{3-\delta}$ and $Ce_{0.8}Sm_{0.2}O_{1.9}$ are highly dispersed and where submicrometer-scale particles form conductive networks with broad pore channels. Such structure is realized by fabricating the electrode structure from the raw powder material stage using spray pyrolysis. The solid oxide electrolysis cells with the nanocomposite electrodes exhibit high current density in steam electrolysis operation (e.g., at 1.3 V), reaching 3.13 A cm$^{-2}$ at 750 °C and 4.08 A cm$^{-2}$ at 800 °C, corresponding to a hydrogen production rate of 1.31 and 1.71 L h$^{-1}$ cm$^{-2}$ respectively.

[1] Inorganic Functional Materials Research Institute, Department of Materials and Chemistry, National Institute of Advanced Industrial Science and Technology (AIST), 2266-98 Anagahora, Shimo-shidami, Moriyama-ku, Nagoya, Aichi 463-8560, Japan. [2] Research Institute for Energy Conservation, Department of Energy and Environment, National Institute of Advanced Industrial Science and Technology (AIST), 1-1-1 Higashi, Tsukuba, Ibaraki 305-8565, Japan. *email: h.shimada@aist.go.jp

nergy is an essential component for individuals and society. Global warming is now regarded as a crucial issue worldwide, and thus each country has been required to reduce greenhouse gas emissions related to energy consumption. Renewable energy sources such as photovoltaic and wind power have thus gained even more attention and have rapidly spread worldwide[1,2]. Because the electric power output of renewable energy sources depends on local climate, widespread use of these resources leads to a gap between supply and demand of electric power, resulting in an increase in surplus electric power. To attain a sustainable society in the future, electrolysis systems are a candidate technology for storage of such surplus electric power due to their high efficiency in converting electric power to chemical energy carriers[3–5].

Low temperature electrolyzers (<100 °C) for hydrogen production using alkaline or proton exchange membranes are already commercially available, reaching an energy-conversion efficiency of approximately 80%[6–8]. Recently, high-temperature solid oxide electrolysis cells (SOECs) have been actively investigated as a next generation electrolysis system that can achieve even higher energy-conversion efficiency (≈100%)[9–14]. High-temperature operation (≥600 °C) enables SOECs to efficiently use input electric power and also waste heat from other systems. SOECs have additional advantages such as low electrode overpotential[15,16], catalysts that are free of noble metals such as platinum and ruthenium[15,16], and various energy carrier production systems via steam and $CO_2$ co-electrolysis[17–20]. Furthermore, SOECs can be used for reversible solid oxide fuel cells (SOFCs) that in a single device can operate both in fuel cell mode to generate electric power and in electrolysis mode to produce chemical energy carriers. In terms of current density, however, further high value is crucial to minimize cell-stack and system size of SOECs and thus reduce their capital cost. Although recent SOECs have demonstrated high current densities of approximately 1 A cm$^{-2}$[9–14], rapid and extensive commercialization of SOECs requires a much higher current density (>3 A cm$^{-2}$) that exceeds that of any existing state-of-the-art electrolyzers including low temperature types.

A critical issue for increasing the current density of SOECs is oxygen electrode performance, which is determined by two factors, namely, intrinsic material properties and electrode structure. Although the intrinsic material properties such as electrochemical catalytic activity and electrical conductivity are of primary importance[21–28], the actual oxygen electrode performance strongly depends on the electrode structure[29–34]. In addition to using high catalytic and high conductive materials for oxygen electrodes, achieving an extremely high current density requires a finely controlled electrode structure, where the electrochemically active triple-phase boundary (TPB) region among the electronic, ionic, and gas phases is expanded as much as possible while maintaining the electronic, ionic, and pore channels. A proposed electrode structure includes a composite oxygen electrode consisting of an electronic conductive perovskite oxide and an ionic conductive oxide. Moreover, because a composite of two materials can effectively improve the long-term durability of oxygen electrodes by preventing sintering of either oxide during high-temperature operation, composite oxygen electrodes are widely applied in current SOECs. For example, the most commonly used oxygen electrode material is a composite of a perovskite oxide of $(La,Sr)(Co,Fe)O_3$ (LSCF) and a $CeO_2$-based oxide in a wide temperature range (550–800 °C). Another perovskite oxide of $(La,Sr)MnO_3$ (LSM) is often used in a composite electrode with a $ZrO_2$-based oxide such as yttria-stabilized zirconia (YSZ) at high temperatures (800–1000 °C)[35–40].

In the present study, to achieve extremely high current density operation over 3 A cm$^{-2}$ in SOECs, we focus on improving the oxygen electrode performance by controlling both the material chemical composition and electrode structure. For the electronic conductive and ionic conductive phases of the electrodes, we respectively use a perovskite oxide of $Sm_{0.5}Sr_{0.5}CoO_{3-\delta}$ (strontium-doped samarium cobaltite, abbreviated as SSC), whose electronic conductivity and catalytic activity are higher than both LSCF and LSM[24–27], and a fluorite oxide of $Ce_{0.8}Sm_{0.2}O_{1.9}$ (samarium-doped ceria, abbreviated as SDC). The proposed electrodes have nanometer-scale structure where both SSC and SDC nano-size crystallites are highly distributed, and are hereafter called "nanocomposite electrodes". To realize these nanocomposite electrodes, we fabricate the electrode structure from the raw powder material stage via spray pyrolysis to control the individual particle structure at nanometer-scale, thus yielding nanocomposite particles[41–45]. Then, we evaluate the electrochemical performance of SOECs with these SSC-SDC nanocomposite electrodes in steam electrolysis, and finally demonstrate that high current density operation is achieved.

## Results

**Concept of nanocomposite electrodes.** Figure 1 shows schematics of the fabrication of the electrode structure in which the raw powder material structure can be controlled to realize high-performance nanocomposite electrodes. Although the final electrode structure is affected by other fabrication parameters such as mixing materials and sintering temperature, the raw powder material is the most important factor. We therefore first prepared the SSC-SDC nanocomposite particles by using spray pyrolysis because the material composition and structure can be controlled by changing the precursor type and/or process parameters[41–45]. Figure 1a shows the spray pyrolysis apparatus used in this fabrication, in which particles were continually prepared via the following steps: (i) atomization of droplets, (ii) transport of droplets and synthesis of particles, and finally (iii) capture of the particles. Figure 1b shows an illustration of a representative SSC-SDC nanocomposite particle prepared by spray pyrolysis. In general, a nanocomposite particle prepared by spray pyrolysis is a submicron-size secondary particle (0.1–1 μm) formed with nano-size crystallites (1–20 nm). Simultaneously synthesized crystallites such as SSC and SDC are highly dispersed within a particle. Note that to strictly control the chemical composition of the targeted materials in a simultaneous synthesis process via spray pyrolysis, the number of cations contained in the target materials needs to be as few as possible. Because the chemical composition of SSC and SDC selected in the present study is composed of only four cations, namely, Co, Sr, Ce, and Sm, the SSC-SDC nanocomposite particles are a very suitable material for spray pyrolysis. As shown in Fig. 1c, the prepared SSC-SDC nanocomposite particles were then used to fabricate oxygen electrodes, namely, SSC-SDC nanocomposite electrodes. The SSC and SDC nano-size crystallites contained in each particle yield a large surface area and a large TPB region within the electrodes. Furthermore, the submicron-size secondary particles become interconnected during the electrode sintering process, thus forming well-connected uniform networks of both SSC and SDC while maintaining the pore channels. The use of SSC-SDC nanocomposite particles prepared by spray pyrolysis enables control of the electrode structure at two phases, i.e., nanometer-scale and submicrometer-scale, resulting in bimodal-structured SSC-SDC nanocomposite electrodes.

**Characteristics of SSC-SDC nanocomposite particles.** In the present study, five different types of SSC-SDC nanocomposite particles with different SSC:SDC composition ratio were prepared by spray pyrolysis. The designed SSC:SDC weight ratios were

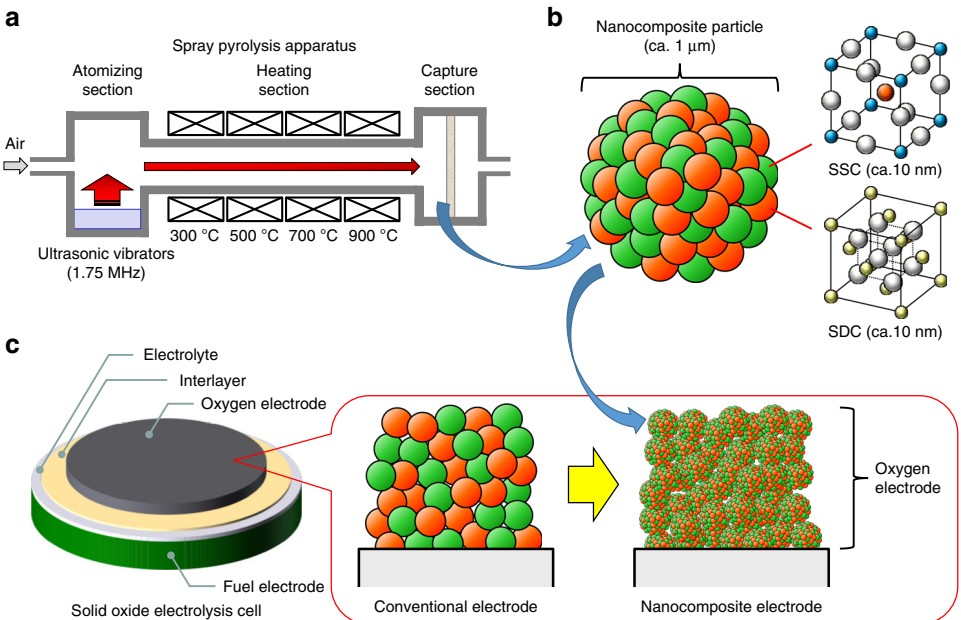

**Fig. 1** Schematics of nanocomposite electrodes. **a** Spray pyrolysis apparatus consisting of atomization, heating, and capture sections. **b** Conceptual images of an SSC-SDC nanocomposite particle and **c** a bimodal-structured SSC-SDC nanocomposite electrode fabricated using the SSC-SDC nanocomposite particles.

80:20, 70:30, 60:40, 50:50, and 40:60, respectively, denoted here as SSC-SDC(80:20), SSC-SDC(70:30), SSC-SDC(60:40), SSC-SDC(50:50), and SSC-SDC(40:60). Figure 2 shows structural characterization results of these prepared SSC-SDC nanocomposite particles. The particles in field emission scanning electron microscopy (FE-SEM) images (Fig. 2a and Supplementary Fig. 1) are secondary particles formed with nano-size crystallites. All the as-prepared SSC-SDC nanocomposite particles had spherical morphology with smooth surface structure regardless of their SSC:SDC composition ratio. Also, the particle size distribution was very narrow for all the SSC-SDC nanocomposite particles (Table 1 and Supplementary Fig. 2), indicating that our preparation process can successfully control the size of the secondary particles. For a representative sample of SSC-SDC(50:50) nanocomposite particles, the particle diameter at 10%, 50%, and 90% of cumulative less-than volume distribution (denoted as $D_{10}$, $D_{50}$, and $D_{90}$) was 0.62, 0.92, and 1.30 μm, respectively. Achieving uniform particle size is essential to realize fine structure electrodes because that leads to a uniform and well-connected network structure within the finally obtained electrodes fabricated using the particles as starting material. The results show sufficiently narrow particle size distributions to successfully construct the bimodal-structured SSC-SDC nanocomposite electrode.

The X-ray diffraction (XRD) patterns for the as-prepared SSC-SDC nanocomposite particles (Fig. 2b) show broad but distinct peaks, all of which correspond to SSC or SDC, without any impurity phases. At the as-prepared condition, SSC and SDC were synthesized as nano-size crystallites in each particle; the estimated crystal size (based on the Scherrer equation) ranged from 13 to 15 nm for SSC and 14 to 19 nm for SDC (Table 1). After the SSC-SDC nanocomposite particles were sintered at 950 °C for 1 h in air (Supplementary Fig. 3), the peaks became sharper and thus more clearly identifiable as SSC and SDC. In all the SSC-SDC nanocomposite particles, the estimated crystal size was approximately 40 nm for both SSC and SDC (Supplementary Table 1), indicating that although the SSC and SDC crystallites increased in size during the sintering, they remained nano-order

in size. This resulting crystal size can be expected to be maintained during SOEC operation because SOECs are generally operated between 600 and 800 °C, which is sufficiently lower than the sintering temperature of 950 °C of the SSC-SDC nanocomposite particles. Also, the Rietveld refinement was carried out for the XRD results of the sintered SSC-SDC nanocomposite particles to estimate the SSC:SDC composition ratio. The resulting SSC:SDC weight ratio was 80:20, 70:30, 59:41, 48:52, and 38:62, respectively, for SSC-SDC(80:20), SSC-SDC(70:30), SSC-SDC(60:40), SSC-SDC(50:50), and SSC-SDC(40:60). These values were almost the same as the target weight ratio, indicating that the SSC:SDC composition ratio could be controlled for all the prepared nanocomposite particles.

The inner structure of the SSC-SDC nanocomposite particles was revealed by using transmission electron microscopy (TEM) and scanning transmission electron microscopy (STEM). Figure 2c shows the cross-section of a representative SSC-SDC (50:50) nanocomposite particle. The particle had a dense inner structure clearly formed with fine crystallites approximately 10–30 nm in size, which corresponds relatively well to the sizes estimated from the XRD patterns and the Scherrer equation (Table 1). Furthermore, energy dispersive X-ray spectroscopy (EDX) results in Fig. 2d reveal that all the constituent elements (Co, Sr, Ce, Sm, and O) of an SSC-SDC(50:50) nanocomposite particle were evenly dispersed and overlapped each other over the entire region of the particle. Spray pyrolysis can produce various types of particle structure, e.g., hollow, porous, and dense[41–45]. To use a raw powder material for SOEC electrodes, a dense structure with evenly dispersed nano-size crystallites is optimal because such structure generally leads to a large TPB region and well-connected networks in the resulting electrodes. In summary, we successfully determined the crucial properties of the raw powder materials for fabricating high-performance SSC-SDC nanocomposite electrodes, namely, spherical, uniform-sized secondary particles, accurate chemical composition and composite ratio, dense inner particle structure, and highly dispersed nano-size crystallites.

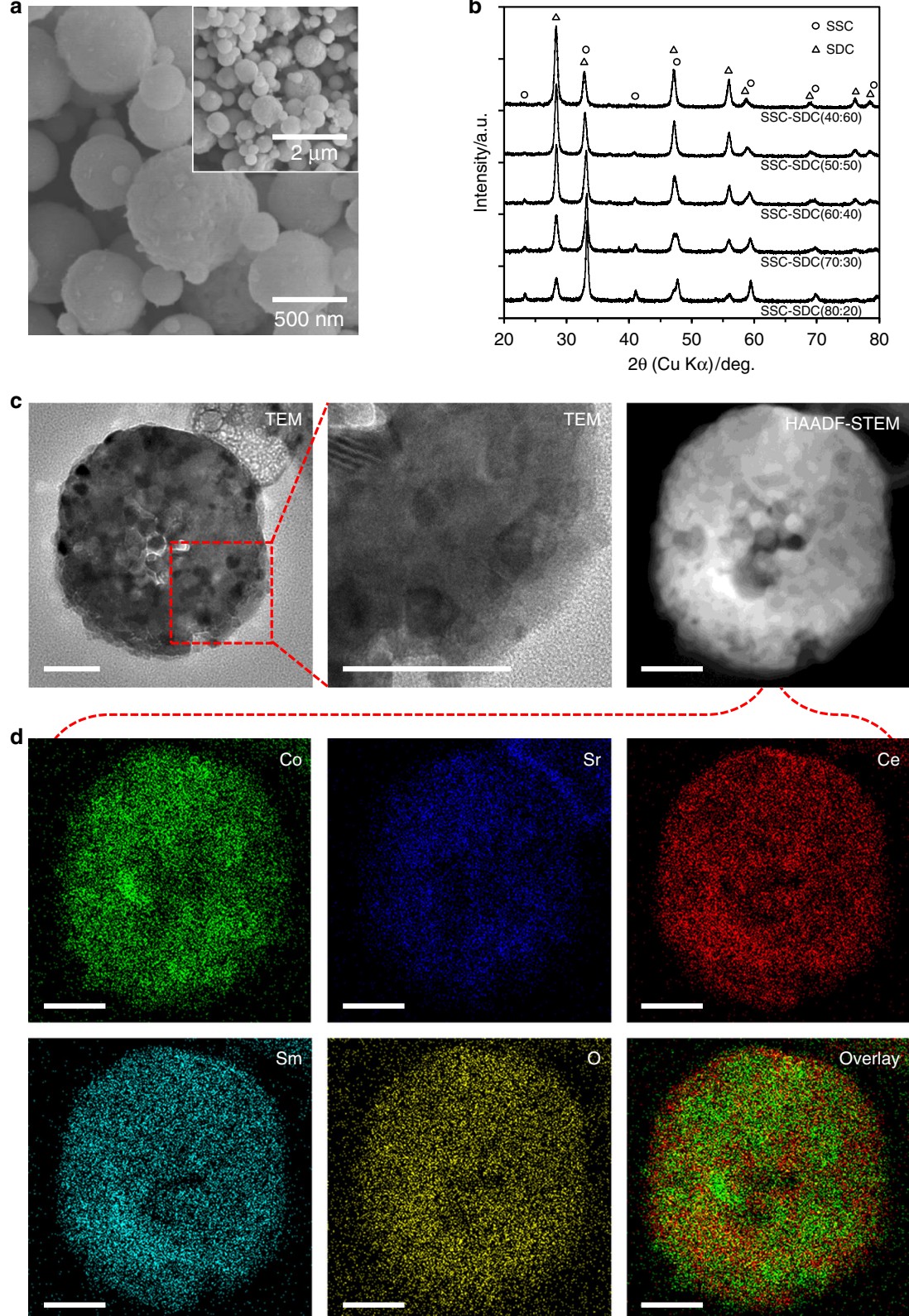

**Fig. 2** Characterization of nanocomposite particles prepared by spray pyrolysis. **a** FE-SEM images of a representative SSC-SDC(50:50) nanocomposite particles. **b** XRD patterns for as-prepared SSC-SDC nanocomposite particles. **c** TEM and HAADF-STEM images and **d** EDX mappings of Co, Sr, Ce, Sm, and O of a representative SSC-SDC(50:50) nanocomposite particle. Overlay image of Co and Ce mapping is also shown. Scale bars in **c** and **d** indicate 100 nm.

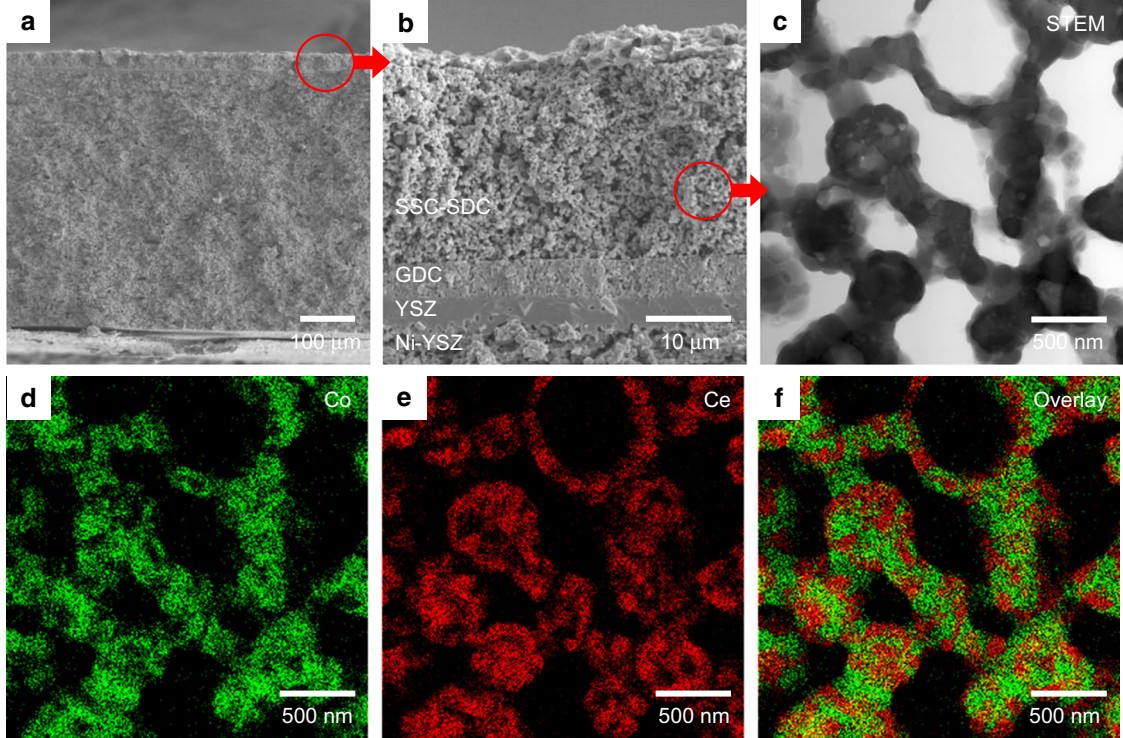

**Fig. 3** Cross-sectional images of solid oxide electrolysis cell with nanocomposite electrode. **a** FE-SEM image of the SOEC consisting of four components, i.e., SSC-SDC(50:50) nanocomposite electrode, GDC interlayer, YSZ electrolyte, and Ni-YSZ fuel electrode. **b** Enlarged FE-SEM image of interfacial structure between each component. **c** STEM image, **d**, **e** EDX mappings of Co and Ce, and **f** overlay image of Co and Ce mappings of SSC-SDC(50:50) nanocomposite electrode.

**Table 1 Crystal size and particle size of nanocomposite particles.**

| Sample | Crystal size (nm) | | Particle size (µm) | | | SSA (m² g⁻¹) |
|---|---|---|---|---|---|---|
| | SSC | SDC | $D_{10}$ | $D_{50}$ | $D_{90}$ | |
| SSC-SDC(80:20) | 13 | 19 | 0.70 | 1.06 | 1.68 | 10.0 |
| SSC-SDC(70:30) | 13 | 17 | 0.75 | 1.07 | 1.55 | 10.3 |
| SSC-SDC(60:40) | 15 | 15 | 0.75 | 1.07 | 1.50 | 12.0 |
| SSC-SDC(50:50) | 15 | 14 | 0.62 | 0.92 | 1.30 | 9.5 |
| SSC-SDC(40:60) | 15 | 15 | 0.59 | 0.88 | 1.27 | 8.4 |

*SSA specific surface area measured by Brunauer-Emmett-Teller (BET) method*

**Microstructure of SSC-SDC nanocomposite electrodes**. Planar SOEC samples with SSC-SDC nanocomposite electrodes were fabricated using the SSC-SDC nanocomposite particles as the raw powder material for oxygen electrodes. To achieve high current density operation, we selected a fuel electrode support configuration that enables the electrolyte thickness to be as thin as a single micron. Figure 3 shows cross-sectional images of a representative SOEC with SSC-SDC(50:50) nanocomposite electrode. The FE-SEM images (Fig. 3a, b and Supplementary Figs. 4–8) reveal that the SOECs were composed of four component layers, namely, SSC-SDC nanocomposite electrode (approximately 20 µm thick), $Ce_{0.9}Gd_{0.1}O_{1.95}$ (GDC) interlayer (≈5 µm), YSZ (8 mol% $Y_2O_3$) electrolyte (≈5 µm), and Ni-YSZ fuel electrode (≈0.55 mm). Note that the NiO in the fuel electrodes was reduced to Ni because the FE-SEM observation was carried out for the SOEC samples after electrochemical measurements in which $H_2$

was the steam carrier. In SOEC operation, current density is often limited by steam diffusion in a fuel electrode[10]. The Ni-YSZ fuel electrode in the present study was therefore designed to have high porosity and large pore size by optimizing the extrusion process[46], and the resulting porosity was 50% and the median pore diameter was 0.93 µm calculated based on volume (Supplementary Fig. 9). The YSZ electrolyte deposited on the porous Ni-YSZ fuel electrode was entirely densified without any defects, suggesting that it can prevent cross-leakage of gas. The GDC interlayer acts as a blocking layer against formation of high resistive phases, such as $SrZrO_3$ and $Sm_{0.2}Zr_{0.2}O_7$, between the SSC-SDC and YSZ[24,47]. Then, as the top layer of the SOECs, the porous, fine-structured SSC-SDC nanocomposite electrode was deposited as the oxygen electrode on the GDC interlayer.

The nanostructure of SSC-SDC nanocomposite electrodes was revealed in more detail by using STEM and EDX. As shown in Fig. 3c, SSC-SDC(50:50) nanocomposite electrode formed with fine crystal grains, of whose apparent size was 10–100 nm. Figure 3d, e show the EDX mappings of Co and Ce, respectively, indicating the SSC and SDC phases. The EDX mappings of Sr overlapped that of Co, and that of Sm and O overlapped that of both Co and Ce (Supplementary Fig. 10). The overlay image of the Co and Ce mappings (Fig. 3f) clearly shows that SSC and SDC phases were fully separated and highly dispersed with respective to each other, thus yielding an extensive TPB with gas phase. STEM and EDX observation was also conducted for another representative sample of SSC-SSD(70:30) nanocomposite electrode (Supplementary Fig. 11) and a structure similar to SSC-SSD (50:50) nanocomposite electrode. Because spherical, uniform-sized particles were used as the starting material, the SSC and SDC grains provided well-connected electronic and ionic conductive networks with sufficiently broad pore channels to

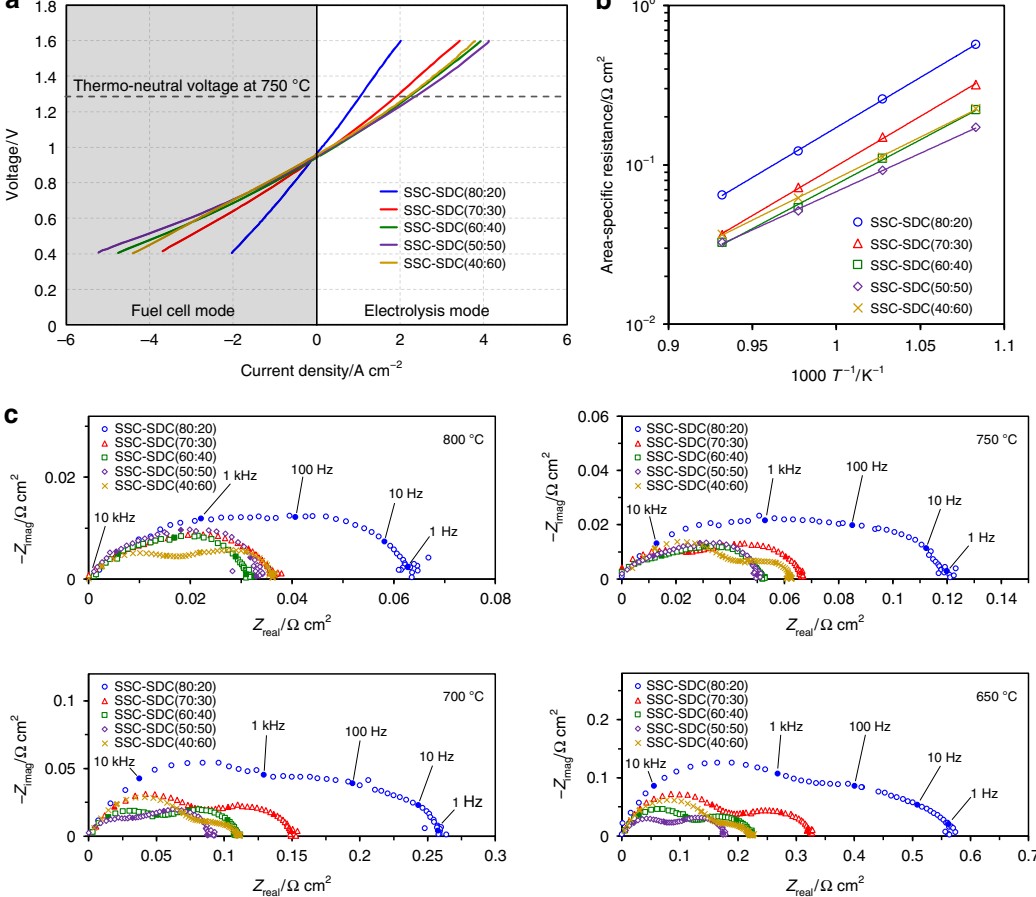

**Fig. 4** Comparison in electrochemical performance of nanocomposite electrodes. SOECs with the SSC-SDC nanocomposite electrodes where SSC:SDC composition ratio was 80:20–40:60 were operated under 40% humidity condition. **a** I–V characteristics in both electrolysis and fuel cell modes at 750 °C. **b** ASRs of electrode polarization and **c** impedance spectra at 650–800 °C under OCV condition.

promote gas diffusion. This network structure leads to high electronic and ionic conductivities and to further three-dimensional extension of the electrochemically active TPB region. These results suggest successful fabrication of bimodal-structured SSC-SDC nanocomposite electrodes for high current density operation in high-temperature electrolysis.

**Performance of SOECs with SSC-SDC nanocomposite electrodes.** The SOECs with the SSC-SDC nanocomposite electrodes were operated using $H_2O$–$H_2$ gaseous mixture fed to the fuel electrode and using air fed to the oxygen electrode. Figure 4a shows the current–voltage (I–V) characteristics at 750 °C under 40% relative humidity of steam using $H_2$ as the steam carrier in both electrolysis and fuel cell modes, namely, reversible SOEC–SOFC operation. In electrolysis mode, all five SOECs exhibited good cell performance; their current densities at 1.3 V, which is close to the thermo-neutral voltage of steam electrolysis (1.28 V), were higher than 1 A cm$^{-2}$. Comparing the SSC-SDC nanocomposite electrodes, the cell performance clearly increased with increasing SDC composition ratio up to 50 wt%, and then that decreased at 60 wt%. The current density at 1.3 V of the SOEC with SSC-SDC(50:50) nanocomposite electrode reached 2.46 A cm$^{-2}$, which is two times higher than that with SSC-SDC (80:20) nanocomposite electrode. Similar tendency, such as high performance and dependence of the current density on the SDC composition ratio, was also confirmed in fuel cell mode, indicating the proposed nanocomposite electrodes can be applied to SOFC and reversible SOEC–SOFC applications.

To evaluate the electrode performance of the five SOECs with the SSC-SDC nanocomposite electrodes, their area-specific resistance (ASR) of electrode polarization was measured using electrochemical impedance spectroscopy (EIS) in a temperature range of 650 to 800 °C. Our primary purpose of the ASR and EIS measurements was to determine the effect of nanocomposite on the electrode performance by comparing electrodes with different SSC:SDC composition ratio. Although EIS under operating conditions is useful for the detailed analysis of current and/or voltage dependence for a cell sample, EIS under OCV condition is generally suitable to compare electrode performance (Supplementary Table 2). We therefore evaluated EIS under OCV condition here. Figure 4b shows ASR as a function of temperature and Fig. 4c shows the ohmic-free impedance spectra at each temperature. The ASR was determined from the difference between the two intercepts on the x-axis at the lower and higher frequency sides of the obtained impedance spectra. The ASR obtained here involves both oxygen electrode and fuel electrode resistances because the proper separation between these two electrode resistances is difficult to determine in electrode-supported cell configurations where the reference electrode cannot be applied[48]. However, differences in the ASR of each SOEC can be attributed to the SSC-SDC nanocomposite electrodes because all the SOECs had identically designed fuel electrodes. As shown in Fig. 4b, the ASR depended on the level of SDC in the SSC:SDC composition ratio, and the SOEC with SSC-SDC(50:50) nanocomposite electrode exhibited the lowest ASR among the five samples. For example, the ASR at 750 °C was 122,

72, 54, 52, and 62 mΩ cm$^2$ for SSC-SDC(80:20), SSC-SDC(70:30), SSC-SDC(60:40), SSC-SDC(50:50), and SSC-SDC(40:60) nano-composite electrodes, respectively. These results correspond to the I–V characteristics (Fig. 4a), indicating the measured high current density in the I–V characteristics was due to high performance of SSC-SDC(50:50) nanocomposite electrode. Both I–V and ASR measurements at 20% humidity showed similar results to those at 40% humidity (Supplementary Figs. 12–14). In general, for an oxygen electrode, SSC can be used as single phase material without being a composite of any other material because SSC intrinsically possesses some ionic conductivity with electro-nic conductivity, namely, mixed ionic and electronic conductiv-ities (MIEC). Actually, SSC as a single phase material is often used in SOEC applications, although in some cases a small amount of CeO$_2$-based oxides such as SDC and GDC is added to SSC to match the thermal-expansion coefficient of the oxygen electrode with that of the other components[25–27]. Alternatively, recent studies have reported a possibility that the TPB region plays an important role in the electrochemical reaction in MIEC-based electrodes[49]. In the present study, the increase in SDC in the composition ratio enhanced the oxygen electrode perfor-mance, suggesting that our nanocomposite technique to produce a large, active TPB region is effective for achieving higher performance even in MIEC materials such as SSC.

Figure 5 and Supplementary Fig. 15 show the measured electrochemical performance of the SOEC with SSC-SDC(50:50) nanocomposite electrode that demonstrated the highest perfor-mance among our five nanocomposite electrodes. As shown in Fig. 5a, the current density of this SOEC under 50% humidity condition using a gaseous mixture of H$_2$ and N$_2$ (10:90 vol%) as steam carrier was further increased to 3.13 A cm$^{-2}$ at 750 °C. To the best of our knowledge, this current density is the highest value among reported data for high-temperature steam electrolysis under similar conditions (Supplementary Table 3). Furthermore, by increasing the temperature from 750 to 800 °C, the SOEC subsequently achieved a high current density of 4.08 A cm$^{-2}$ at 1.3 V. Also, even at a lower temperature of 650 °C, the current density of the SOEC was approximately 1 A cm$^{-2}$ at 1.3 V. These outstanding high current densities are considered to be due to a synergy effect between the highly active material and the nanocomposite structure applied to the proposed electrode. The intrinsically high catalytic activity of SSC was further enhanced by introducing SDC, and the nanocomposite structure broadly extended the active TPB region. The corresponding hydrogen production rates calculated using the Faraday's raw at 750 and 800 °C were 1.31 and 1.71 L h$^{-1}$ cm$^{-2}$, respectively. Note that the current efficiency of 100% was assumed here in estimating the hydrogen production rates because YSZ is an absolute ionic conductive electrolyte without electronic conduction. Also, the OCV of the SOEC at each condition was measured to confirm no current leakage and no gas leakage in our experiments. For example, the OCV under 50% humidity condition at 750 °C was 0.854 V, which was comparable to the theoretical OCV calculated using the Nernst equation (0.856 V). Higher production rate leads to an increase in the amount of output products from an electrolyzer system. These results therefore suggest that SSC-SDC (50:50) nanocomposite electrode enables SOECs to produce a large amount of hydrogen due to significantly high current density in a wide temperature range. Moreover, in cases of steam and CO$_2$ co-electrolysis, the proposed electrode would contribute to a high production rate of syngas (H$_2$ and CO).

Besides high current density, high durability is essentially needed for oxygen electrode materials in SOECs. In general, cobaltite-based oxides including SSC have high sinterability and exhibit instability such as Sr diffusion. In practice, SSC-based electrodes are often used below 700 °C to prevent such

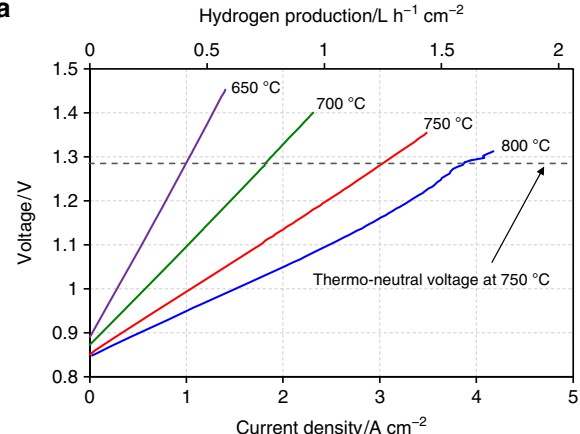

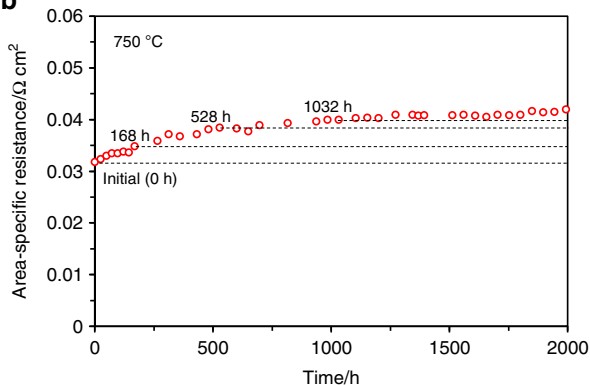

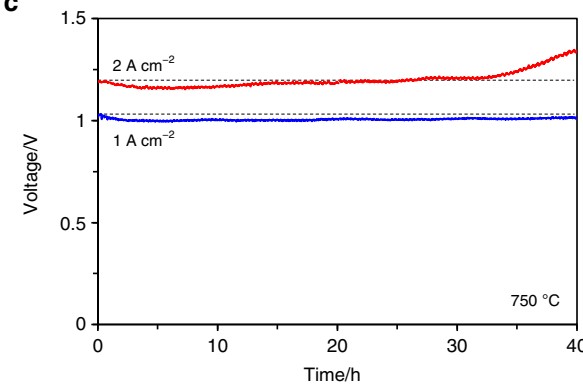

**Fig. 5** Electrochemical performance of optimized nanocomposite electrode. The highest performance SSC-SDC(50:50) nanocomposite electrode among our nanocomposite electrodes was evaluated. **a** I–V characteristics and corresponding hydrogen production for the SOEC at 650–800 °C under 50% humidity condition. **b** ASR of electrode polarization using a symmetrical cell at 750 °C in air under open-circuit condition as a function of operating time for 2000 h. **c** Short-term durability for the SOEC under electrolysis operating condition. Time-dependent voltage was measured at 750 °C under 40% humidity condition at constant current densities of 1 and 2 A cm$^{-2}$.

degradation (Supplementary Table 4)[50–56]. However, the single material SSC is chemically stable under oxygen electrode side atmosphere even at much higher temperatures. For example, the basic properties such as the conductivity and thermal expansion of SSC have been stably measured at high temperatures above 800 °C[27,57–60]. Also, SSC raw powder materials are often synthesized via high-temperature process above 1000 °C[61–64]. Consequently, the degradation of SSC-based electrodes is not due

to essential chemical stability of the single material SSC, but rather is due to high sinterability and incompatibility with the electrolyte material such as YSZ in cases in which a porous electrode of an SOEC is used. We therefore examined the following durability tests.

As mentioned above, SSC intrinsically has high sinterability. In addition, due to the high surface area, nano-size materials generally undergo grain growth at high temperatures where SOECs and SOFCs can operate. Thus, the long-term durability of SSC-SDC(50:50) nanocomposite electrode against degradation caused by sintering during high-temperature operation was evaluated here. In this evaluation, a symmetrical cell with a GDC electrolyte disk was used to measure the ASR that was due only to SSC-SDC(50:50) nanocomposite electrode and not to the fuel electrode. Figure 5b shows the time-dependent ASR at 750 °C for 2000 h in air. As expected, the initial ASR was significantly low, 32 mΩ cm². In the initial period of approximately 500 h, the ASR increased; for example, increased to 35 mΩ cm² at 168 h and to 38 mΩ cm² at 528 h. The increase rate decreased with increasing operating time, until the ASR stabilized after 1000 h. Although the ASR increased during the initial period, the actual ASR value for SSC-SDC(50:50) nanocomposite electrode was very low, ≈40 mΩ cm² for the entire 2000 h even at a high operating temperature of 750 °C. The main reason for this low ASR is that the highly dispersed SSC and SDC prevented grain growth of either component. Another contributing factor might be the structural networks of electronic, ionic, and gas phases, which formed due to uniform submicron-size secondary particles. To investigate grain growth behavior in more detail, we carried out high-temperature XRD analysis for SSC-SDC(50:50) nanocomposite electrode at 750 °C for 70 h in air (Supplementary Fig. 16). The XRD patterns remained unchanged during the measurements, indicating that SSC and SDC remained at their initial crystal size (approximately 40 nm). As shown here, the high-temperature XRD analysis revealed no change in the crystal sizes, whereas the ASR in the initial period exhibited some degradation. We therefore consider that the ASR degradation was caused by a mechanism other than sintering, for example, a mechanism related to the chemical composition of SSC and SDC.

As demonstrated here, the proposed nanocomposite structure enabled the electrode to achieve good long-term durability against degradation caused by sintering as well as to achieve high performance. Electrode performance was successfully maintained for 2000 h using the GDC electrolyte disk without electric bias, although some initial degradation occurred. The second reason for the degradation of SSC-based electrodes is chemical incompatibility with $ZrO_2$-based electrolytes, namely, forming $SrZrO_3$ resistive phase at high temperatures (>700 °C)[24,47]. We therefore applied a GDC interlayer, which is a well-known technique to prevent formation of $SrZrO_3$ resistive phase. Also, because the current density shown in the present study is much higher than previously reported values, such high current density operation might affect the durability of an SOEC. Figure 5c shows the results of short-term durability tests under electrolysis operating conditions at constant current densities of 1 and 2 A cm⁻² using the SOEC with SSC-SDC(50:50) nanocomposite electrode. During the measurements, 40% humidified $H_2$–$N_2$ gaseous mixture (10:90 vol%) was fed to the fuel electrode and air was fed to the oxygen electrode. Results reveal that the voltage of the SOEC under 1 A cm⁻² operation kept the initial value for 40 h, whereas that under 2 A cm⁻² operation increased after 10 h depending on time, indicating degradation. Because the SOEC deteriorated when a constant current density of 2 A cm⁻² was applied, durability tests at further higher current densities were not performed in the present study. A possible reason for the degradation at a high current density of 2 A cm⁻² is formation of

$SrZrO_3$ at the interface between the GDC interlayer and the YSZ electrolyte. Although the GDC interlayer was used as a blocking layer against $SrZrO_3$ formation in our SOECs, the interlayer could not completely suppress such formation probably due to its porous structure. In future work, we will therefore develop SOECs with a perfectly dense interlayer and clarify the long-term durability of the nanocomposite electrodes under aggressive high current density operation.

## Discussion

Nanocomposite electrodes consisting of a perovskite oxide of SSC and a fluorite oxide of SDC were proposed as high-performance oxygen electrodes to achieve higher current density exceeding 3 A cm⁻² in SOECs. To realize an extremely fine electrode structure, our nanocomposite electrodes were designed from the raw powder material stage. First, SSC-SDC nanocomposite particles that were secondary particles formed with SSC and SDC nano-size crystallites were prepared by spray pyrolysis and then were applied as an electrode material. The resulting nanocomposite electrodes had a bimodal structure where nanometer-scale SSC and SDC extended the TPB region, and submicrometer-scale particles constructed both electronic and ionic conductive networks with broad pore channels. The SOECs applied with the SSC-SDC nanocomposite electrodes exhibited excellent cell performance. By using the optimized SSC-SDC nanocomposite electrode, the current density at 1.3 V reached 3.13 A cm⁻² at 750 °C and 4.08 A cm⁻² at 800 °C in steam electrolysis, both values of which are significantly higher that of conventional SOECs (approximately 1 A cm⁻² at 800 °C). The corresponding hydrogen production rates at 750 and 800 °C were, respectively, 1.31 and 1.71 L h⁻¹ cm⁻², thus enabling reduction in both the size and cost of the electrolyzer while maintaining a high energy-conversion efficiency. The present high-performance electrodes will contribute various applications needed in a sustainable society in the future, such as energy storage that utilizes renewable energy, transport of energy carriers, and electric power generation via SOEC systems and reversible SOEC–SOFC systems.

## Methods

**Synthesis of SSC-SDC nanocomposite particles**. Five $Sm_{0.5}Sr_{0.5}CoO_{3−δ}$ (SSC) and $Ce_{0.8}Sm_{0.2}O_{1.9}$ (SDC) nanocomposite particles with different SSC:SDC composition ratio (80:20, 70:30, 60:40, 50:50, and 40:60 by weight) were synthesized by spray pyrolysis. These nanocomposite particles are respectively denoted as SSC-SDC(80:20), SSC-SDC(70:30), SSC-SDC(60:40), SSC-SDC(50:50), and SSC-SDC(40:60) nanocomposite particles. The spray pyrolysis apparatus consisted of an atomization section, a heating section, and a capture section (Fig. 1a). First, five nitrate solutions for spray pyrolysis were prepared by dissolving of $Co(NO_3)_2·6H_2O$ (98.0% purity, Kanto Chemical Co.), $Sr(NO_3)_2$ (98.0% purity, Kanto Chemical Co.), $Ce(NO_3)_3·6H_2O$ (98.5% purity, Kanto Chemical Co.), and $Sm(NO_3)_3·6H_2O$ (99.95% purity, Kanto Chemical Co.) in distilled water. The SSC-SDC total cation concentration of each nitrate solution was adjusted to 0.1 mol L⁻¹. The amount of actually used nitrates is described in Supplementary Table 5. Then, the nitrate solutions were atomized by three ultrasonic vibrators (1.75 MHz) in the atomization section. The atomized droplets of each nitrate solution were transported by air as carrier gas at a flow rate of 3 L min⁻¹ through a quartz tube (1.5-m long and 52-mm inner diameter) to the heating section, where four electric furnaces at 300, 500, 700, and 900 °C were serially set respectively from inlet to outlet. In the heating section, the droplets were continuously dried, dehydrated, reacted, and finally crystallized. The residence time of droplets or reactants in each electric furnace was approximately 13 s. The synthesized SSC-SDC nanocomposite particles were captured by a polytetrafluoroethylene filter in the capture section.

**Fabrication of SOECs with SSC-SDC nanocomposite electrodes**. To prepare the fuel electrode, NiO powder (Sumitomo Metal Mining Co.), YSZ powder (TZ-8YS, Tosoh), graphite carbon (UF-G10, Showa Denko), and cellulose (TG-101, Asahi Kasei Chemicals) at a weight ratio of 60:40:23:11 was mixed with binder, and then stirred in a vacuum chamber with adding a proper amount of distilled water. By aging the powder mixture in ambient atmosphere for 15 h, NiO-YSZ clay for extrusion was obtained. The clay was extruded using a screw type extruder (Miyazaki Iron Works Co.) with a metal mold (0.7-mm thick, 120-mm wide) to make a NiO-YSZ green sheet. The detailed fabrication procedure of the extrusion

process has been presented elsewhere[46]. Button-size pieces (32-mm diameter) were cut from the green sheet and then sintered at 1240 °C for 2 h in air, producing pre-sintered NiO-YSZ fuel electrode substrates. An YSZ paste for the electrolyte layer was prepared by mixing YSZ powder (TZ-8Y, Tosoh) in α-terpineol (Kanto Chemical Co.) with ethyl cellulose (45 cP, Kishida Chemical Co.), a dispersant, and a plasticizer. The YSZ paste was screen-printed on the pre-sintered NiO-YSZ fuel electrode substrates and then was co-sintered at 1360 °C for 3 h in air, resulting in a dense YSZ electrolyte and porous NiO-YSZ fuel electrode assembly. A GDC paste for the interlayer was prepared by mixing GDC powder ($Ce_{0.9}Gd_{0.1}O_{1.95}$, Solvay Special Chem Japan) and the same admixtures as in the YSZ paste. The GDC paste was screen-printed on the dense YSZ electrolyte layer and then sintered at 1300 °C for 1 h in air. Five SSC-SDC pastes of different SSC:SDC composition ratio were prepared for the oxygen electrode using the SSC-SDC nanocomposite particles with the same admixtures as in the YSZ and GDC pastes. Each SSC-SDC paste was then screen-printed on the GDC interlayer, and finally sintered at 950 °C for 1 h in air. The effective area of each oxygen electrode was 0.283 $cm^2$. The obtained oxygen electrodes are denoted as SSC-SDC(80:20), SSC-SDC(70:30), SSC-SDC(60:40), SSC-SDC(50:50), and SSC-SDC(40:60) nanocomposite electrodes, respectively.

**Fabrication of symmetrical cells.** Dense GDC disks were prepared using the same GDC powder as the interlayer of the SOEC samples. The disks (26-mm diameter) were formed by uniaxially pressing the GDC powder at 50 MPa for 1 min and then sintering at 1350 °C for 3 h in air. After sintering, the obtained dense GDC disks were approximately 20-mm diameter and 1-mm thick. The SSC-SDC paste for preparing SSC-SDC(50:50) nanocomposite electrode was screen-printed on both sides of the GDC disk symmetrically. Finally, the symmetrical cell consisting of the GDC disk and SSC-SDC(50:50) oxygen electrode (0.283-$cm^2$ effective area and approximately 20-μm thick) was obtained by sintering at 950 °C for 1 h in air.

**Characterization.** The crystal structure of the as-prepared and sintered SSC-SDC nanocomposite particles was determined using XRD (SmartLab, Rigaku) at room temperature with Cu Kα radiation. Also, high-temperature XRD (X'Pert Pro MPD, PANalytical) was carried out for the SSC-SDC nanocomposite electrode sample at 750 °C with Cu Kα radiation. The crystal sizes of SSC and SDC were estimated using the Scherrer equation with a Scherrer constant of 0.9. The morphology of the prepared samples was observed by using FE-SEM (JSM-6330F, JEOL) and TEM/STEM (JEM-2100F, JEOL). The atomic concentration of each contained element in the samples, namely, Co (Kα, 6.929 keV), Sr (Lα, 1.806 keV), Ce (Lα, 4.839 keV), Sm (Lα, 5.635 keV), and O (Kα, 0.523 keV), was analyzed using EDX (JED-2300T, JEOL) equipped with TEM/STEM. To measure the particle size distribution of the SSC-SDC nanocomposite particles, each particle sample was dispersed in distilled water with sodium hexametaphosphate, and then the size distribution was measured using a laser scattering particle size analyzer (Microtrack HRA 9320-X100, Nikkiso Co.). The specific surface area was measured by the BET method with $N_2$ as the adsorption species using a gas sorption analyzer (Autosorb-iQ, Quantachrome Instruments). For the Ni-YSZ fuel electrode, the porosity was determined by the Archimedes method and the pore size by mercury porosimetry.

**Electrochemical measurements.** The I–V and EIS measurements for the SOECs were carried out using an electrochemical interface (1287, Solartron) and a frequency response analyzer (1255B, Solartron). During these measurements, the SOECs were operated at different temperatures from 650 to 800 °C with gaseous mixtures of $H_2O$ in $H_2$ and $N_2$ (through a water bubbler) at an $H_2$ and $N_2$ total flow rate of 100 mL $min^{-1}$ fed to the fuel electrode and with air at a flow rate of 100 mL $min^{-1}$ fed to the oxygen electrode. EIS data were obtained in the frequency range of 1 MHz to 0.1 Hz and with an amplitude of applied voltage of 10 mV. To reduce current collection loss, Pt paste (<1 μm thick) was painted onto the top surface of both the oxygen and fuel electrodes. The thermocouple to measure the operating temperature was placed close (approximately 1 mm) to the oxygen electrode surface of the SOEC sample.

## Data availability

The data that support the findings of this study are available from the corresponding author upon reasonable request.

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

## Acknowledgements

This work was partially supported by the Core Research for Evolutionary Science and Technology program of the Japan Science and Technology Agency (JST CREST Grant Number JPMJCR1343).

## Author contributions

H.S. carried out the sample preparation, electrochemical measurements, and characterization. T.Y. and Y.F. planned the present work. H.K. advised in the interpretation of all the data. H.Sumi supported the electrochemical measurements. Y.Y. and K.N. supported the particle characterization. H.S. prepared and edited the manuscript with input from all the authors.

## Competing interests

The authors declare no competing interests.
