## [Peer Review File · Nature Communications]

Reviewers' comments:

Reviewer #1 (Remarks to the Author):

This manuscript reports a work on developing a nanocomposite electrode of SSC-SDC for solid oxide electrolysis cells with impressive high current density. The work is interesting; however, the overall novelty and data quality for this work cannot meet the requirements for publishing in this prestigious journal. I would suggest for a more specific journal.

The specific comments are as following.

1. The language throughout the manuscript needs significant improvement as a professional paper.
2. SSC stability is questionable at such a high temperature. Actually SSC is a well know cathode material for reduced temperature SOFC while it has exhibited chemical instability in 700oC and above in literature. This is why it is barely reported in YSZ based cells and widely used for GDC or SDC based cells.
3. 500-nm particles clearly are not nano range. As a result, it is appropriate to claim nano-composite.
4. Figure 1, 2, and 3 should be included into supporting material as they provide only general information for readers to confirm the particle size, distribution and microstructure of button cells. The main figures should focus on delivering the most important and relevant results.
5. It is very ambiguous to claim the highest performance in state-of-art high- and low-temperature SOECs. To support the claim with sufficient evidence, the conclusion should be supported by a comprehensive comparison figure/table for SOECs (O₂- SOEC, H-SOEC) at similar conditions (temperature, humidity, electrode material, voltage, etc.).
6. In Fig. 4c, the impedance in OCV conditions should not be used for discussion as it cannot represents the real operating condition in SOEC with applied voltage. The electrode polarization under voltage is totally different from those in OCV conditions.
7. Fig. 5a, did author measure hydrogen production rate, such as observing hydrogen concentration change with gas chromatography/MS? It is not correct to directly convert current density to hydrogen production rate without confirming Faradaic efficiency.
8. In Fig. 5b, electrode stability should be evaluated at real operating conditions in full cells. Because the high current density can cause degradation problem. Again, the impedance should be measured with voltage bias.
9. The HT-XRD result in 70h was used to claim crystal size not changed which cannot convince no change in particle size. The 2000h ASR results exactly show the degradation process. Therefore, these two results conflict with each other.

Reviewer #2 (Remarks to the Author):

In this manuscript, nanocomposite electrodes consisting of a perovskite oxide of SSC and a fluorite oxide of SDC were proposed as high-performance oxygen electrodes to achieve higher current density over 3 A cm⁻² in SOECs. The nanocomposite electrode is interesting, and the achieved performance is impressive. The experiment method and procedure are new and innovative, and the data are reliable. However, there is unclear mechanism proposed for understanding. It should be publishable after addressing the following comments:

1. In the introduction section, some important and the most updated literature should be cited, like: Electrochemical Energy Reviews 2018, 1, 433-459, Electrochemical Energy Reviews 2018, 1, 54-83;
2. What is the energy-conversion efficiency of the SOECs.
3. The authors claimed that " the increase in composition ratio of SDC enhanced the oxygen electrode performance." Does the performance always increase as the increase of SDC? Currently, the maximum ratio of the designed SDC: SSC was 50:50, Therefore, the electrochemical performance for

the samples obtained at several higher composition ratio of SDC should be provided to reveal which one is the best material in electrochemical performance and further analyze the reasons in detail.

4. As shown in Fig. 3c, SSC-SDC(50:50) nanocomposite electrode formed with fine crystal grains of approximately 10-100 nm. How do the authors get the sizes (10-100 nm) ?.

5. In Line 268-269, Page 15. The authors claimed that "the obtained current density of 2.46 A cm⁻² is the highest value among already reported data of steam electrolysis under similar conditions." Please indicate the reported data or list the relevant literatures.

6. In Line 322-325, Page 18. The authors claimed that "These results suggest that SSC-SDC(50:50) nanocomposite electrode enables SOECs to produce a large amount of hydrogen and/or other energy carriers due to significantly high current density in a wide temperature range." What is the reason that the SSC-SDC(50:50) nanocomposite electrode enables SOECs to produce a large amount of hydrogen and/or other energy carriers? Authors should give further explanation about this phenomenon.

7. In Line 338-339, Page 18. The authors claimed that "the reason for this low ASR is that the highly dispersed SSC and SDC prevented grain growth of either component." The authors simply attributed the dispersion of SSC and SDC without further evidence. The particle size, or other factors may also be beneficial for this low ASR.

8. In Line 343-347, Page 19. The authors claimed that "as demonstrated here, the proposed nanocomposite structure enabled the electrode to achieve good long-term durability against degradation caused by sintering as well as to achieve high performance for high current density. Because the current density shown in the present study is much higher than all the previously reported values....." Please list the reported values and make a detailed comparison and discussion. Move importantly, please give a clearly convincing mechanism to explain why the SOECs with the proposed SSC-SDC nanocomposite structure can achieve the significantly high current density in steam electrolysis operation, namely, at 1.3 V, reaching 3.13 A cm⁻².

Reviewer #3 (Remarks to the Author):

The authors reported an interesting achievement of >3 A/cm² current density in a solid oxide electrolysis cell, which is a significant advancement in developing next generation electrolysis system for ~100% energy conversion efficiency. They focused on improving the oxygen electrode performance by optimizing chemical composition for lowering kinetic barrier and electrode structure for minimize electronic, ionic, gas transport resistances. The design is logically clear, and the results are mostly convincing. I recommend publication after minor revision and believe their approaches from design to catalyst synthesis to electrode fabrication are interesting to the researchers in the field. Below is questions and suggestions for the authors:

Why SDC, not CS, is used for Ce_{0.8}Sm_{0.2}O_{1.9} (SDC), while SSC is used for Sm_{0.5}Sr_{0.5}CoO_{3-δ} (SSC)?

Why not further decrease the SSC:SDC ratio when the lowest is the best?

In the abstract, "further high current density, ...is essential to ..." may be change to "further increasing current density ..."

Do "nano-order level" and "submicron-order level" mean nanometer-scale and submicrometer-scale?

Response to the reviewers:

We gratefully appreciate the reviewers for spending immense time on our manuscript. Our manuscript has been revised in consideration of their valuable comments. All parts revised according to the reviewers' comments are highlighted in red in the revised manuscript and supplemental information files.

To Reviewer #1

Reviewer #1 (Remarks to the Author):

This manuscript reports a work on developing a nanocomposite electrode of SSC-SDC for solid oxide electrolysis cells with impressive high current density. The work is interesting; however, the overall novelty and data quality for this work cannot meet the requirements for publishing in this prestigious journal. I would suggest for a more specific journal.

The specific comments are as following.

We owe a debt of gratitude to Reviewer #1 for invaluable advice. We revised our manuscript as follows:

C1. The language throughout the manuscript needs significant improvement as a professional paper.

A1. We checked the language of the entire manuscript ourselves and also asked a native speaker who has extensive experience editing scientific papers.

C2. SSC stability is questionable at such a high temperature. Actually SSC is a well know cathode material for reduced temperature SOFC while it has exhibited chemical instability in 700°C and above in literature. This is why it is barely reported in YSZ based cells and widely used for GDC or SDC based cells.

A2. In the previous version of our manuscript, description about the proper operating temperature for SSC was insufficient, despite its importance. As stated in your comment, SSC is generally used below 700 °C because sintering easily progresses at temperatures above 700 °C. This was one of the challenges in our work. By controlling the nanostructure of SSC-contained electrodes, our SSC-SDC nanocomposite electrode exhibited good sintering durability in tests using a GDC disk

at open-circuit (without bias) even at 750 °C. Also, our XRD results (Fig. 2b) show that SSC has good chemical compatibility with ceria-based oxides such as SDC. A critical issue is instability between SSC and YSZ, which is the material we used for the electrolyte in our present study. Although we therefore added a GDC interlayer to our SOECs, this interlayer was not fully dense, which led to degradation in voltage during the electrolysis test at 2 A cm⁻² using a single SOEC (Supplementary Fig. 17). In future work, we intend to investigate SOECs with a fully dense interlayer. We added a description of the test results for the higher operating temperatures. (pages 19, 20).

Supplementary Figure 17

- C3. 500-nm particles clearly are not nano range. As a result, it is appropriate to claim nanocomposite.
- A3. The 500-1000 nm particles prepared by spray pyrolysis were secondary particles consisting of approximately 10-nm SSC and SDC crystallites. Also, the grain size of the electrode prepared using these particles was approximately 40 nm. Thus, we call these particles “nanocomposite particles” and call the electrode a “nanocomposite electrode” (page 8).
- C4. Figure 1, 2, and 3 should be included into supporting material as they provide only general information for readers to confirm the particle size, distribution and microstructure of button cells. The main figures should focus on delivering the most important and relevant results.

-
- A4. Thank you for your comment. Of course, we agree that key figures should focus on delivering the most important and relevant results. We believe that Figs 1, 2, and 3 are needed in the main manuscript for the following reasons. First, Fig. 1 shows the concept and scheme of the present study so that readers readily understand our research results. Top journals, including *Nature Communications*, often show concept and/or schematics of the research in the first figure of an article. Second, Figs. 2 and 3 show key results in our study and are not general information. Although the essential results in this study are the high electrochemical performance of the nanocomposite electrodes, the nanostructure and microstructure are critical factors in determining the performance. Our challenge is to achieve such nanocomposite structure shown in Figs. 2 and 3. Thus, we show particle size, distribution, and structure results as key figures.
- C5. It is very ambiguous to claim the highest performance in state-of-art high- and low temperature SOECs. To support the claim with sufficient evidence, the conclusion should be supported by a comprehensive comparison figure/table for SOECs (O2-SOEC, H-SOEC) at similar conditions (temperature, humidity, electrode material, voltage, etc.).
- A5. Based on your advice, we added a table for comprehensive comparison in the supplementary information file (Supplementary Table 2) and added a description about this comparison in the revised manuscript (page 18). Because our SOECs are oxide ion-conducting type using YSZ electrolyte, Supplementary Table 2 summarizes high-temperature oxide ion-conducting SOECs.
- C6. In Fig. 4c, the impedance in OCV conditions should not be used for discussion as it cannot represent the real operating condition in SOEC with applied voltage. The electrode polarization under voltage is totally different from those in OCV conditions.
- A6. We agree with your comment, and we understand the difference in EIS results between OCV and operating conditions. Per your advice, we added EIS results for the nanocomposite electrode that had the highest performance, namely, SSC-SDC(50:50) (Supplementary Fig. 15). We think that this is a critical but difficult issue for comparison of electrode performance. For example, at 1.3 V operating condition, the EIS results for all SOEC samples were affected by the gas diffusion process because our SOEC could operate at very high current density. Also, even at 1.3 V, which is near thermo-neutral voltage, the actual temperature might be different for each oxygen electrode. Our primary purpose of the EIS measurements was to determine the

effect of nanocomposite on the electrode performance by comparing electrodes with different SSC:SDC composite ratio. We therefore show the EIS results at the OCV condition in the main manuscript.

Supplementary Figure 15

C7. Fig. 5a, did author measure hydrogen production rate, such as observing hydrogen concentration change with gas chromatography/MS? It is not correct to directly convert current density to hydrogen production rate without confirming Faradaic efficiency.

A7. In our study, hydrogen production rate was calculated from current density by using Faraday's law and assuming 100% current efficiency. The electrolyte material of our SOECs was YSZ, which has absolute oxide-ionic transport property without electronic conduction. To check gas leakage, such as cross-leakage of the electrolyte and gas-sealing, we measured the OCV for each gas condition. In the revised manuscript, we added a description of a representative measured OCV (page 18).

C8. In Fig. 5b, electrode stability should be evaluated at real operating conditions in full cells. Because the high current density can cause degradation problem. Again, the impedance should be measured with voltage bias.

A8. Based on your advice, we added EIS results with voltage bias as described in A6 (Supplementary Fig. 15). Stability during high current density operation is a critical issue in the development of SOEC electrodes. Although many researchers have recently investigated this issue, results that demonstrate no degradation have not yet been reported. During high current density, our electrodes also exhibited some degradation that could not yet be completely overcome. We therefore added the data

for short-term SOEC operation and added a description of future study of in durability of these SOECs (page 20; Supplementary Fig. 17).

C9. The HT-XRD result in 70h was used to claim crystal size not changed which cannot convince no change in particle size. The 2000h ASR results exactly show the degradation process. Therefore, these two results conflict with each other.

A9. As you stated, the HT-XRD results showed no change in particle size and 2000 h ASR results showed degradation. These results seem to be in conflict. However, HT-XRD can detect only particle size, suggesting ASR degradation was caused by a different mechanism. Although we have not yet obtained definitive evidence, we believe that the degradation was due to a change in the chemical composition of SSC. In the revised manuscript, we added a description about this possible explanation for the degradation (page 19). Currently, we are trying to clarify the mechanism behind this electrode degradation by using various analyses such as SIMS and high-resolution TEM, and expect to report the details on the degradation mechanism, and in future work to finally overcome this electrode degradation.

To Reviewer #2

Reviewer #2 (Remarks to the Author):

In this manuscript, nanocomposite electrodes consisting of a perovskite oxide of SSC and a fluorite oxide of SDC were proposed as high-performance oxygen electrodes to achieve higher current density over 3 A cm⁻² in SOECs. The nanocomposite electrode is interesting, and the achieved performance is impressive. The experiment method and procedure are new and innovative, and the data are reliable. However, there is unclear mechanism proposed for understanding. It should be publishable after addressing the following comments:

We gratefully appreciate the invaluable advice from Reviewer #2 to enhance the quality of our manuscript. We revised our manuscript as follows:

C1. In the introduction section, some important and the most updated literature should be cited, like: Electrochemical Energy Reviews 2018, 1, 433-459, Electrochemical Energy Reviews 2018, 1, 54-83;

A1. Based on your suggestion, we added references to updated literature; “Electrochemical Energy Reviews 2018, 1, 433-459, Electrochemical Energy Reviews 2018, 1, 54-83” (page 4; Refs. 28 and 34).

C2. What is the energy-conversion efficiency of the SOECs.

A2. We assumed 100% energy-conversion efficiency at thermo-neutral voltage because our SOEC used YSZ electrolyte, which is a material that intrinsically has no current-leakage. Also, we calculated the hydrogen production rate from the current density using Faraday’s law and assuming 100% current efficiency. By measuring the OCV, we verified that there was no gas-leakage, such as electrolyte cross-leakage and glass-sealing. In the revised manuscript, we added a description of the measured OCV (page 18).

C3. The authors claimed that " the increase in composition ratio of SDC enhanced the oxygen electrode performance." Does the performance always increase as the increase of SDC? Currently, the maximum ratio of the designed SDC: SSC was 50:50, Therefore, the electrochemical performance for the samples obtained at several higher composition ratio of SDC should be provided to reveal which one is the best material in electrochemical performance and further analyze the reasons in detail.

A3. Based on your suggestion, we carried out additional preparation and evaluation of the higher composition ratio sample with 60 wt% SDC, denoted as SSC-SDC(40:60). Results showed that the nanocomposite electrode with 50 wt% SDC exhibited the highest performance. In the revised manuscript and supplementary information, we report the I-V and EIS measurement results and the characterizations for SSC-SDC(40:60) (Figs. 2, 4; Table 2; Supplementary Figs. 1, 2, 3, 8, 12, 13, 14; Supplementary Table 1). For example, Fig. 2b and 4 are representatively shown below.

Figure 2b

Figure 4

C4. As shown in Fig. 3c, SSC-SDC(50:50) nanocomposite electrode formed with fine crystal grains of approximately 10-100 nm. How do the authors get the sizes (10-100 nm) ?.

A4. The crystal grain size of the nanocomposite electrode was apparently determined from STEM-EDX images (page 12). This nanocomposite electrode structure is essential to achieve high current density. As shown in Fig. 1, to ensure fine crystal grains after high-temperature sintering of the oxygen electrode, we used the SSC-SDC nanocomposite particles for the raw powder material to fabricate the electrodes. Experimental results confirm that grain growth was successfully prevented by the highly dispersed SSC and SDC. The average size of crystal grains estimated from the XRD results using the Scherrer equation was 40 nm for both SSC and SDC. The STEM-EDX results coincide with the XRD results

C5. In Line 268-269, Page 15. The authors claimed that "the obtained current density of 2.46 A cm⁻² is the highest value among already reported data of steam electrolysis under similar conditions." Please indicate the reported data or list the relevant literatures.

A5. Based on your suggestion, in the revised manuscript, we added a table that summarizes previously reported data for steam electrolysis under similar conditions to those in our study. In our previous manuscript, we claimed similar results for two of the humidity conditions, namely, 40% (page 15) and 50% (page 17). In the revised manuscript, we therefore added a description about comparison at only the 50% humidity condition where we report higher current density (page 18; Supplementary Table 2).

C6. In Line 322-325, Page 18. The authors claimed that "These results suggest that SSC-SDC (50:50) nanocomposite electrode enables SOECs to produce a large amount of hydrogen and/or other energy carriers due to significantly high current density in a wide temperature range." What is the reason that the SSC-SDC(50:50) nanocomposite electrode enables SOECs to produce a large amount of hydrogen and/or other energy carriers? Authors should give further explanation about this phenomenon.

A6. Thank you for your advice about further explanation of production rate. Higher production rate indicates a larger amount of output products from an electrolyzer system. In the present study, we experimentally reported only steam electrolysis. In

the revised manuscript, we therefore added a description of suggested co-electrolysis that can produce syngas as another energy carrier (page 18).

C7. In Line 338-339, Page 18. The authors claimed that "the reason for this low ASR is that the highly dispersed SSC and SDC prevented grain growth of either component." The authors simply attributed the dispersion of SSC and SDC without further evidence. The particle size, or other factors may also be beneficial for this low ASR.

A7. In our previously submitted manuscript, we discussed only dispersion of SSC and SDC. As you stated, many factors might affect the resulting ASR. In the revised manuscript, we therefore added a description of other possible factors that might contribute to low ASR (page 19).

C8. In Line 343-347, Page 19. The authors claimed that "as demonstrated here, the proposed nanocomposite structure enabled the electrode to achieve good long-term durability against degradation caused by sintering as well as to achieve high performance for high current density. Because the current density shown in the present study is much higher than all the previously reported values....." Please list the reported values and make a detailed comparison and discussion. Move importantly, please give a clearly convincing mechanism to explain why the SOECs with the proposed SSC-SDC nanocomposite structure can achieve the significantly high current density in steam electrolysis operation, namely, at 1.3 V, reaching 3.13 A cm⁻².

A8. We again appreciate your advice. In the revised manuscript, we compared our results to previously reported data in Supplementary Table 2, and we added a discussion and a reference about why the present nanocomposite electrode effectively achieved a high current density (page 16, 18; Ref. 48).

To Reviewer #3

Reviewer #3 (Remarks to the Author):

The authors reported an interesting achievement of >3 A/cm² current density in a solid oxide electrolysis cell, which is a significant advancement in developing next generation electrolysis system for ~100% energy conversion efficiency. They focused on improving the oxygen electrode performance by optimizing chemical composition for lowering

kinetic barrier and electrode structure for minimize electronic, ionic, gas transport resistances. The design is logically clear, and the results are mostly convincing. I recommend publication after minor revision and believe their approaches from design to catalyst synthesis to electrode fabrication are interesting to the researchers in the field. Below is questions and suggestions for the authors:

We express our gratitude for the time spent and valuable advice given by Reviewer #3 to improve the manuscript and also to encourage our research. We revised our manuscript as follows:

C1. Why SDC, not CS, is used for $\text{Ce}_{0.8}\text{Sm}_{0.2}\text{O}_{1.9}$ (SDC), while SSC is used for $\text{Sm}_{0.5}\text{Sr}_{0.5}\text{CoO}_{3-\delta}$ (SSC)?

A1. SDC stands for Sm-doped CeO_2 . Although “CSO” is sometimes the abbreviation for $\text{Ce}_{0.8}\text{Sm}_{0.2}\text{O}_{1.9}$, “SDC” is the most popular abbreviation. We therefore use “SDC”. Because we inadvertently missed defining the abbreviation “SDC” in the previously submitted manuscript, we added the definition in the revised manuscript (page 5).

C2. Why not further decrease the SSC:SDC ratio when the lowest is the best?

A2. Thank you for the advice. In the revised manuscript, we added the characteristics and electrochemical results of a higher composition ratio of SDC (60 wt%) to clearly show the best composition ratio among the SSC-SDC nanocomposite electrodes. As a result, we concluded that the composition ratio of SSC:SDC = 50:50 was the best (Figs. 2, 4; Table 2; Supplementary Figs. 1, 2, 3, 8, 12, 13, 14; Supplementary Table 1).

C3. In the abstract, “further high current density, ...is essential to ...” may be change to “further increasing current density ...”

A3. Per your suggestion. we revised the sentence in the abstract (page 2).

C4. Do “nano-order level” and “submicron-order level” mean nanometer-scale and submicrometer-scale?

A4. Yes. For clarity, we therefore respectively changed “nano-order level” and “submicron-order level” to “nanometer-scale” and “submicrometer-scale” throughout the revised manuscript.

Reviewers' comments:

Reviewer #1 (Remarks to the Author):

The responses didn't well address our questions and concerns, and the revised manuscript didn't satisfy me at all, which are reflected in the following aspects:

1. SSC showed intrinsic instability when operated at high temperatures, which has been reported in a lot of SOFC work. It does not make any sense to me show a durable operation under more aggressive condition (SOEC). The authors seem to misunderstand my question.
2. The author has no intent to fully address my suggestion on EIS result. All EIS shown in main text are still under OCV, which are misleading.
3. Stability test on symmetrical cells does not necessarily indicate SOEC's durability, which could partially validate my hypothesis: SSC is not stable at all at this temperature.
4. The point the author tried to sell is high performance. However, a simple IV characteristics plot is insufficient to justify the feasibility of the concept.

Overall, I think the author's efforts in the revision and response was not spot on. Due to the inherent drawback of the material system author used at inappropriate temperatures, and experiment design, data acquisition and analysis they presented, I don't recommend it for publication in Nature Communication.

Reviewer #2 (Remarks to the Author):

I am satisfied with the revision. No more revision is further needed.

Response to the reviewer:

We gratefully appreciate the editor and reviewers for spending immense time on our manuscript. Our manuscript has been revised in consideration of their valuable comments. All parts revised according to the reviewer's comments are highlighted in red in the revised manuscript and supplemental information files.

Reviewer's comment 1:

SSC showed intrinsic instability when operated at high temperatures, which has been reported in a lot of SOFC work. It does not make any sense to me show a durable operation under more aggressive condition (SOEC). The authors seem to misunderstand my question.

Our response 1:

As stated in the comment, SSC ($\text{Sm}_{0.5}\text{Sr}_{0.5}\text{CoO}_3$) electrodes used in SOFCs exhibit instability in high temperature operation above 700°C. However, the chemical stability of the single material SSC is high under oxygen electrode condition even at much higher temperatures, e.g., 800-1000°C. Many researchers have confirmed the stability of the intrinsic properties such as conductivity and thermal expansion of SSC at high temperatures above 800°C (page 19). Furthermore, in preparation of SSC raw powder material, the synthesis temperature is much higher than 800°C (page 19). In the case of present study, the synthesis temperature is 900°C, and many previously reported studies have shown synthesis temperatures higher than 1000 °C. These are evidences for the stability of the single material SSC.

Consequently, the instability of SSC is not due to any intrinsic property of the single material SSC, but rather is due to two factors: sinterability and compatibility in cases in which a porous electrode of a solid oxide cell is used. In the first factor, the degradation of SSC electrodes is related to changes in the microstructure caused by intrinsic high sinterability. In our study, improvement in sintering durability at 750 °C operation due to nanocomposite structure was confirmed by high-temperature XRD (Supplementary Figure 16) and 2000 h symmetrical cell tests (Figure 5b). Operation at 750°C is a challenge but it is never impossible because the SSC electrodes already have been applied to SOECs and SOFCs at a temperature around 700 °C (Supplementary Table 4).

The second factor is chemical incompatibility with zirconia-based electrolytes. At high

temperature ($>$ about $700\text{ }^{\circ}\text{C}$), SSC forms SrZrO_3 resistive phase with zirconia. We therefore applied a GDC (Gd-doped ceria) interlayer, which is a well-known technique to prevent SrZrO_3 resistive phase, and was found effective in the present study. Although operation at 2 A/cm^2 high current density exhibited some degradation, 1 A/cm^2 current density operation is stable in 40 h short-term tests (Supplementary Figure 17). The degradation of 2 A/cm^2 high current density operation was caused by the GDC interlayer not completely suppressing SrZrO_3 formation due to its porous GDC interlayer structure (page 20).

In addition, the reason why many researchers have used SSC at lower temperatures ($<$ $700\text{ }^{\circ}\text{C}$) is related to the electrolyte material. SSC is often used with ceria-based electrolytes (not zirconia) due to good chemical compatibility. Because ceria-based electrolytes show electron conductivity (current leakage) above $650\text{ }^{\circ}\text{C}$, these types of solid oxide cells with SSC electrodes have been operated below $650\text{ }^{\circ}\text{C}$ (Supplementary Table 4). In the cases using other electrolytes, such as zirconia-based electrolytes with GDC interlayer and $(\text{LaSr})(\text{GaMn})\text{O}_3$ -based electrolytes, SSC electrodes have been successfully operated at around $700\text{ }^{\circ}\text{C}$ (Supplementary Table 4).

Reviewer's comment 2:

The author has no intent to fully address my suggestion on EIS result. All EIS shown in main text are still under OCV, which are misleading.

Our response 2:

Although EIS (electrochemical impedance spectrum) under operating condition is important as Reviewer #1 suggested, the basic data of EIS must be that measured under OCV. The case in which EIS under operating condition is needed is, for example, the detailed analysis of current and voltage dependence for a cell. To compare electrode performance, EIS under OCV condition is generally suitable (page 15). Actually, in recent articles about solid oxide cells published in journals of Nature Publishing Group, almost all researchers have shown EIS data under OCV condition (Supplementary Table 2). Thus, we consider that EIS under OCV condition is more proper for our study.

Reviewer's comment 3:

Stability test on symmetrical cells does not necessarily indicate SOEC's durability,

which could partially validate my hypothesis: SSC is not stable at all at this temperature.

Our response 3:

As described above in “Our response 1”, a main reason for SSC electrode degradation is sinterability. Thus, the symmetrical cell test is essential to show the sintering durability of our SSC-SDC nanocomposite electrode.

Reviewer’s comment 4:

The point the author tried to sell is high performance. However, a simple IV characteristics plot is insufficient to justify the feasibility of the concept.

Our response 4:

Of course, a simple *I-V* characteristics plot by itself is insufficient for justification of this work. However, the primary evidence must be *I-V* characteristics. We therefore also show EIS and durability of our developed nanocomposite electrodes, as well as issues for future work. This approach is common among studies by top researchers (Ref. 1–6).

References

1. Li, M. et al., A niobium and tantalum co-doped perovskite cathode for solid oxide fuel cells operating below 500 °C. *Nat. commun.* **8**, 13990 (2017).
2. Li, T. et al., Design of next-generation ceramic fuel cells and real-time characterization with synchrotron X-ray diffraction computed tomography. *Nat. commun.* **10**, 1497 (2019).
3. Xia, C. et al., Shaping triple-conducting semiconductor $\text{BaCo}_{0.4}\text{Fe}_{0.4}\text{Zr}_{0.1}\text{Y}_{0.1}\text{O}_{3-\delta}$ into an electrolyte for low-temperature solid oxide fuel cells. *Nat. commun.* **10**, 1707 (2019).
4. Zhou, Y. et al., Strongly correlated perovskite fuel cells. *Nature.* **534**, 231–234 (2016).
5. Duan, C. et al., Highly durable, coking and sulfur tolerant, fuel-flexible protonic ceramic fuel cells. *Nature.* **557**, 217–222 (2018).
6. Choi, S. et al., Exceptional power density and stability at intermediate temperatures in protonic ceramic fuel cells. *Nat. Energy* **3**, 202–210, (2018).

REVIEWERS' COMMENTS:

Reviewer #2 (Remarks to the Author):

1. SSC showed intrinsic instability when operated at high temperatures, which has been reported in a lot of SOFC work. It does not make any sense to me show a durable operation under more aggressive condition (SOEC). The authors seem to misunderstand my question.

My opinion: I think the authors have given a detailed explanation about the stability issue of SSC. There are some differences between SOFC and SOEC. Due to this manuscript is focused on SOEC, testing the performance at aggressive conditions may be needed for demonstrating their SOEC material's durability.

2. The author has no intent to fully address my suggestion on EIS result. All EIS shown in main text are still under OCV, which are misleading.

My opinion: OCV measurement is a portion of the electrochemical characterization. If the authors' experiment conditions allow, measuring the EIS at different operation current densities or voltages is even more convincing. However, in some cases, measure EIS at OCV is good enough for illustration purpose.

3. Stability test on symmetrical cells does not necessarily indicate SOEC's durability, which could partially validate my hypothesis: SSC is not stable at all at this temperature.

My opinion: I think the symmetrical cell measurements should be ok for testing the material's stability.

4. The point the author tried to sell is high performance. However, a simple IV characteristics plot is insufficient to justify the feasibility of the concept.

My opinion: the IV characteristic is normally used to show the performance of a SOFC or SOEC, and to prove the device's feasibility.

Overall, in my opinion, this manuscript should be accepted for publication because it has the novelty and new data.

Response to the reviewer:

We gratefully appreciate again the editor and reviewers for spending immense time on our manuscript. Our manuscript has been revised in consideration of the reviewer's valuable comments. All parts revised according to the reviewer's comments are highlighted in red in the revised manuscript file.

Reviewer's opinion 1:

I think the authors have given a detailed explanation about the stability issue of SSC. There are some differences between SOFC and SOEC. Due to this manuscript is focused on SOEC, testing the performance at aggressive conditions may be needed for demonstrating their SOEC material's durability.

Our response 1:

We moved the results of short-term durability tests under electrolysis operating conditions from Supplementary Information (Supplementary Fig. 17) to the main manuscript (Fig 5c). These testing were carried out with applying current densities of 1 and 2 A cm⁻². Because our SOEC deteriorated when the applied current density was 2 A cm⁻² as shown in Fig. 5c, the durability test at higher current densities was not done in the present study. Although more aggressive condition measurements such as 3 A cm⁻² were not shown in here, 1 and 2 A cm⁻² are much higher current densities compared with the present SOEC level. In future work, we would like to show the degradation mechanism in more detailed with clear evidence, and would like to try again durability test at much higher current density.

Reviewer's opinion 2:

OCV measurement is a portion of the electrochemical characterization. If the authors' experiment conditions allow, measuring the EIS at different operation current densities or voltages is even more convincing. However, in some cases, measure EIS at OCV is good enough for illustration purpose.

Our response 2:

In the present time, we have no more EIS data under operating condition. Thus, we

would like to measure EIS data at various operating conditions in future works which focus on the dependence of the electrode polarization resistance on current and voltage.

Reviewer's opinion 3:

I think the symmetrical cell measurements should be ok for testing the material's stability.

Our response 3:

Thank you for your understanding our data.

Reviewer's opinion 4:

The IV characteristic is normally used to show the performance of a SOFC or SOEC, and to prove the device's feasibility.

Our response 4:

Thank you for your understanding our data.